# Expression of Elafin and CD200 as Immune Checkpoint Molecules Involved in Celiac Disease

**DOI:** 10.3390/ijms25020852

**Published:** 2024-01-10

**Authors:** Candelaria Ponce-de-León, Pedro Lorite, Miguel Ángel López-Casado, Pablo Mora, Teresa Palomeque, María Isabel Torres

**Affiliations:** 1Department of Experimental Biology, Faculty of Health Sciences, University of Jaén, 23071 Jaén, Spain; cponce@ujaen.es (C.P.-d.-L.); plorite@ujaen.es (P.L.); pmora@ujaen.es (P.M.); tpalome@ujaen.es (T.P.); 2Department of Pediatric Gastroenterology, Virgen de las Nieves Hospital, 18014 Granada, Spain; drlopezcasado@digestivointegral.es

**Keywords:** CD200/CD200R, Elafin, celiac disease, gluten peptides, immune checkpoint

## Abstract

We comprehensively evaluated the expression of therapeutically targetable immune checkpoint molecules involved in celiac disease (CD). We have focused on the alteration of the CD200/CD200R pathway and Elafin expression in celiac disease and discussed their roles in regulating the immune response. There are limited data related to the expression or function of these molecules in celiac disease. This finding could significantly contribute to the understanding of the clinical manifestation of CD. CD200, CD200R and Elafin distributions were determined by ELISA and immunohistochemistry analyses in serum and biopsies of CD patients. Analyses of Th1 and Th17 cytokines were determined. PCR amplification of a fragment of the *PI3* gene was carried out using genomic DNA isolated from whole blood samples of the study subjects. Different aliquots of the PCR reaction product were subjected to RFLP analysis for SNP genotyping and detection. We characterized the expression and function of the CD200–CD200R axis and *PI3* in celiac disease. A significantly higher level of soluble CD200 and CD200R and lower expression of *PI3* in serum of CD patients was observed compared to healthy controls. Consistent with our results, CD200 expression is regulated by IFN-gamma. Interaction of CD200/CD200R leads to production of type-Th1 and -Th17 cytokines. Regarding the *PI3* genotype, the CT genotype proportion SNP rs1733103 and the GG genotype SNP rs41282752 were predominant in CD patients. SNP rs1733103 showed a significant association between the SNP variables and CD. In celiac disease the immune checkpoint is compromised or dysregulated, which can contribute to inflammation and the autoimmunity process. The study of these checkpoint points will lead to the development of targeted therapies aimed at restoring immunological balance in CD. Specific coding regions of the *PI3* gene-splice variants predispose the Elafin protein, both at the transcriptional and post-translational levels, to modify its expression and function, resulting in reduced differential functional protein levels in patients with active celiac disease.

## 1. Introduction

Celiac disease (CD) is an autoimmune condition that arises from gluten sensitivity in genetically predisposed individuals [1,2,3]. It is a complex disorder in which genetic, environmental and immunological factors converge [4,5]. Tolerance to dietary antigens is linked to the activation of regulatory T cells and the production of immunosuppressive cytokines, preventing inappropriate responses from effector Th lymphocytes [6]. Gluten consumption by individuals with CD triggers a sequence of inflammatory reactions that result in the characteristic small intestine lesions, leading to partial or total atrophy of the intestinal villi and decreased nutrient absorption [7,8]. Additionally, it activates CD4+ T cells and increases intraepithelial lymphocytes in the intestinal mucosa. Gluten-derived epitopes typically contain multiple proline and glutamine residues that are selectively deamidated by tissue transglutaminase (TG2) [9,10,11]. TG2 is crucial in secondary autoimmunity, generating neo-antigens and triggering autoantibody production in celiac disease. This process leads to the presence of serum autoantibodies, including anti-ganglioside and anti-actin IgA antibodies, in CD patients with neurological disorders or severe mucosal damage [12,13]. TG2’s role sheds light on extraintestinal immune manifestations, revealing the intricate interplay between TG2, autoimmunity and diverse clinical presentations in CD patients. The modified peptides can bind to HLA-DQ2/DQ8, stimulating CD4+ T-helper 1 (Th1) cells in the lamina propria, which, upon recognizing gluten peptides, produce proinflammatory cytokines, primarily interferon gamma (IFN-γ) [14]. Hence, the pathogenic model in CD involves factors acting at both the epithelial and lamina propria levels (Figure 1), including (i) incomplete digestion and trans-epithelial peptide transport, (ii) the impact of gluten on the epithelium, (iii) proliferation and activation of intraepithelial lymphocytes (IELs), and (iv) recognition of gluten peptides by specific T lymphocytes with HLA-DQ2 restriction after modification by tissue transglutaminase (TG2) [15,16,17].

Celiac disease, like other autoimmune diseases, is a polygenic disorder strongly associated with *HLA* genes, particularly the *CELIAC1* locus located on chromosome 6p21. In the majority of celiac patients, a variant of the HLA-DQ2 molecule is present, which is encoded by the *DQA105* and *DQB102* alleles. Other patients carry *DQ8* (*DQA103*, *DQB10302*) alleles or some isolated alleles of *DQ2* [18,19]. These genes exhibit a dose effect mediated by peptide presentation, which is more pronounced in *HLA-DQ2* homozygotes. Interestingly, while approximately 25% of the population carries *DQ2*, only 1% develops CD. These HLA-DQ molecules present deamidated gluten peptides to CD4+ T cells [19]. Whole-genome studies have identified other regions containing susceptibility genes, many of which are related to immune function and potentially shared with other chronic immune-based diseases [20,21,22].

The immune system includes mechanisms that maintain homeostasis, including both activator and inhibitory mechanisms. Immune checkpoints are pivotal in modulating the signaling pathways that are responsible for immunological tolerance. In our research, we have identified alterations in signaling related to the co-inhibitory PD-L1 and the immunosuppressive IDO in celiac disease patients [23,24,25]. We have provided the first evidence of elevated PD-L1 and IDO expression levels in celiac patients on the surface of intestinal epithelial and lamina propria cells, resulting in increased kynurenine levels in the serum of these patients [23,24].

Numerous studies have emphasized the importance of CD200 in controlling autoimmunity and inflammation in diseases associated with heightened immune system activity [26]. The CD200 molecule and its receptor CD200R exert immunomodulatory effects, induce immune tolerance and regulate the differentiation, adhesion and chemotaxis of various cell populations [26]. They also play a crucial role in the release of mediators and cytokines. CD200 modulates macrophage polarization by interacting with CD200R, initiating inhibitory signals that downregulate the pro-inflammatory M1 phenotype and promote the anti-inflammatory M2 phenotype. This balance helps maintain tissue homeostasis. Dysregulation of CD200-CD200R signaling is implicated in inflammatory and autoimmune conditions by disrupting this balance. [27]. While CD200 and CD200R share identical extracellular portions, they differ in their cytosolic COOH tails. CD200 lacks the ability to transmit activation or inhibition signals in its minimal intracellular region, whereas CD200R, upon binding, can downregulate the exaggerated activity of the immune system to protect the organism from damage due to hyperactivity [28,29,30].

Elafin (peptidase inhibitor 3, PI3, and its biologically active precursor, pre-Elafin) is a neutrophil serine proteinase inhibitor with a crucial role in preventing excessive tissue injury during inflammatory events [31]. Elafin is constitutively produced by the epithelium, with its expression regulated by inflammatory stimuli such as proinflammatory cytokines (IL-1b, TNF-a), bacterial lipopolysaccharides and elastase [32]. Specifically, Elafin modulates colon inflammation and possesses antiprotease, immunomodulatory and antiproliferative activities. The Elafin/trappin-2 pathway, with its pleiotropic mechanisms, holds therapeutic promise for intestinal diseases. Functioning as a robust anti-inflammatory agent, Elafin/trappin-2 modulates the inflammatory cascade and plays a vital role in immune response modulation by influencing the function and differentiation of immune cells, contributing to a balanced and regulated immune environment in the intestine [31]. Elafin is a member of the “trappin” gene family and features an amino-terminal transglutaminase substrate domain, identified as a substrate for tissue transglutaminase TG2 cross-linking [31,32]. TG2 plays a critical role in CD pathogenesis by deamidating gluten peptides and increasing the binding affinity of HLA-DQ2 (DQ8) molecules to these peptides [15].

Galipeau et al. [33] explored the role of Elafin in CD and its interaction with factors involved in its pathogenesis by using human small intestinal tissue and in vitro gliadin deamidation assays. They observed a lower presence of the Elafin in patients with active CD, which reflects inflammation and damage to the small intestinal epithelium. Furthermore, they reported an improvement in inflammation and the restoration of the small intestinal epithelial barrier in a gluten-sensitive mouse model treated with Elafin, describing Elafin as a molecule that is capable of interacting with gliadin peptides and affecting TG2, which is involved in their deamidation [33].

Our study thoroughly evaluated the expression of targetable immune checkpoint molecules in CD. Existing data on their expression or function in CD are limited. Our objective was to characterize the expression and function of the CD200/CD200R axis and *PI3* in CD, along with their correlation with other immune checkpoints, such as IDO and PD1/PDL1. Our previous research has associated these checkpoints with CD development. Additionally, our aim was to investigate whether the interaction of CD200/CD200R reduces activation, proliferation and Th17 cytokine production. We hypothesized that specific coding regions of the *PI3* gene splice variants might predispose the Elafin protein to modify its expression and function, resulting in reduced functional protein levels in patients with active CD compared to healthy individuals. These findings could significantly contribute to understanding the clinical manifestation of CD.

## 2. Results

### 2.1. Diagnosis: Serological, Genetic, and Histological Analysis

A total of 62% CD patients were female, and 38% were male. The age range spanned from 1 to 12 years, with 71% of participants falling within the 3 to 12 years range and 29% within the 1 to 2 years range (Appendix A).

Serological analyses revealed positive levels of endomysial antibody (AEMA) and elevated anti-transglutaminase antibody (ATGA) in all included patients diagnosed with CD. None of the patients exhibited IgA deficiency. All patients were confirmed to have active CD. Additionally, peripheral blood samples were collected at the time of diagnosis confirmation, along with small intestinal mucosa biopsies.

Histological analysis was conducted based on the Marsh classification [34]. Intestinal biopsies from CD patients exhibited typical indications of the disease, including villi with total or partial atrophy and increased lymphocyte infiltration. Among the patients were some who had Marsh III lesions (partial or complete villi atrophy and crypt hypertrophy (Appendix A)).

HLA genotype frequency analysis showed that the most common allele among patients was *HLA-DQ2* (*DQA1*0501*, *DQB1*0201*) (76%). The *DQ8* genotype (*DQA1*03:01*, *03:03*–*DQB1*03:02*) was present in 24% of CD patients.

### 2.2. CD200, CD200R and Elafin Distribution in Intestinal Mucosa Tissue from CD Patients by Immunohistochemistry

The diversity in cell types present in intestinal samples made it necessary to develop immunohistochemical methods for of CD200, CD200R and Elafin detection to evaluate the cellular source, in order to provide an adequate interpretation of the results. To achieve this, an immunohistochemistry technique was used. CD200 and CD200R were expressed in epithelial cells in CD, with CD200 showing more intense reactivity compared to the receptor expression (Figure 2A,C). In addition to epithelial expression, CD200- and CD200R-positive cells were also identified in the lamina propria, resembling cells morphologically similar to lymphocytes. Within the Lieberkühn crypts, we found an intense CD200 expression located in the basal part facing the lumen of the crypts, visible both on the cell membrane and as a secreted form within the cytoplasm of the cells (Figure 2B). Also, we found expression on the apical surface of the crypts.

No expression of CD200R was detected at the Lieberkühn crypt level (Figure 2D). However, strong Elafin expression was found in cells of the lamina propria resembling lymphocytes, as well as in epithelial cells in patients with CD (Figure 3). The expression pattern showed expression both in the surface epithelium (Figure 3A,B,D) as well as in crypt cells (Figure 3A,C). 

### 2.3. Detection of CD200, CD200R and Elafin Protein in Serum from CD Patients by ELISA Assays

The potential biomarkers of CD identified in our study, PI3, CD200 and CD200R, were further validated using quantitative ELISA on the same cohort of samples. Our findings revealed higher expression levels of CD200 and CD200R, while PI3 exhibited lower expression levels compared to the healthy control group. As shown in Figure 4, the mean serum levels of CD200 were 32.62 ± 5.03 pg/mL in CD patients and 23.45 ± 0.48 pg/mL in control patients. The mean levels of CD200R were 864 ± 101 pg/mL and 377 ± 66 pg/mL in sera from CD patients and healthy patients, respectively. The mean levels of Elafin were 5217 ± 1543 pg/mL and 7062 ± 1066 pg/mL in sera from CD patients and healthy patients, respectively (Figure 4).

No consistent association was observed between the expression levels of Elafin, CD200 and CD200R with sex, age, clinical stage or histology. However, we assessed the potential correlations between the levels of expression of Elafin, CD200 and CD200R (Figure 4). Our analysis revealed that the levels of CD200 and Elafin exhibited significant correlations that were different from zero, with a confidence level of 95.0% (*p* < 0.05).

### 2.4. CD4 and CD8 IFN-Gamma Expression in CD Patients: Gamma Interferon Responses of CD4 and CD8 T-Cell Subsets

CD4 microbeads were designed for the isolation of human cells based on the CD4 antigen expression. CD8 microbeads were used to isolate highly pure cytotoxic T cells. The relative contributions of CD4 and CD8 T cell subsets to IFN-γ production in CD were examined. Our findings revealed that both CD4+ T cells and CD8+ T cells showed increased IFN-γ secretion in CD compared to control patients (Figure 5).

### 2.5. Analysis of Th1 and Th17 Cytokines

Using immunohistochemistry analysis, we examined the presence of IL-17 and lL-21 in intestinal biopsies obtained from patients with active disease and healthy individuals. The results suggest that IL-21 secretion is likely associated with lamina propria lymphocytes, as there is no evidence indicating that IELs might be also a source of this cytokine (Figure 6B,C). In response to gluten stimulus, IL-21 is produced by lamina propria lymphocytes and it mediates induction of IFN-γ. The distribution of intestinal IL-17+ and IL-22+ cells in the biopsy tissues were immunohistochemically processed and evaluated by confocal microscopy. The results indicate an impaired secretion of both IL-17+ and IL-22+ cells in CD (Figure 6). Also, a significative number of both IL-17- and IL-22-producing cells were found in the small intestinal mucosa.

Previously, a role for IL-23-mediated inflammation in the pathogenesis of CD was demonstrated, showing that pepsin-A/trypsin digest of gliadin (PTG) stimulates the production of IL-23 in PBMC from CD patients [31]. We analyzed the presence of IL-23 in serum from CD patients with active disease. The results revealed a pronounced and statistically significant increase in serum levels of IL-23 in celiac patients compared to healthy individuals (Figure 7).

### 2.6. MIP-4 Protein in CD

An in vitro enzyme-linked immunosorbent assay for the quantitative measurement of human MIP-4 levels in serum was used. The MIP-4 protein is involved in adaptive immunity due to its capability to attract both CD4+ and CD8+ T lymphocytes [32]. Our results demonstrated that macrophages derived from peripheral blood monocytes exhibited elevated levels of MIP-4 protein expression in CD patients (Figure 8).

### 2.7. PI3 Genotype and Frequencies

The allele and genotype frequencies of SNPs rs1733103, rs41282752 and rs2664581 are shown in Table 1. Notably, in this study, the observed genotype frequencies in the control and CD groups were consistent with the Hardy–Weinberg equilibrium.

For the SNP rs1733103, the CC genotype frequency was notably higher in healthy patients compared to celiac patients, where the CT genotype proportion was more prevalent. Statistical analysis using the Pearson’s chi-square analysis revealed a significant association between the SNP variables and the study group (*p* < 0.05). The strength of this association, measured by Cramer’s V, was moderate (0.274). Regarding the genotype association model with the disease risk compared to the reference genotype (CC) between the control and celiac groups, the codominant CT model represented a 2.71-fold increased risk of developing the disease compared to individuals without this genotype, with a statistically significant value (*p* < 0.05) and a goodness of fit (AIC) of 126.6 (Table 2). The dominant CT/TT model represented a 2.00-fold increased risk compared to the reference genotype, with a statistically significant value (*p* < 0.05) and an AIC of 127.3, while the overdominant C/T model represented a 2.60-fold increased risk compared to the reference genotype, with a statistically significant value (*p* < 0.05) and an AIC of 125.4.

For the SNP rs41282752, the GG genotype was more prevalent in the celiac patient group, while the GA genotype was predominant in healthy patients (Table 2). The chi-square analysis indicated no significant relationship between the polymorphism variables and the patient type (*p* > 0.05). Similarly, the Fisher’s exact test also did not yield statistical significance (0.297). Furthermore, no significant associations were found between the patients’ gender and predisposition to CD. Additionally, in the disease development association study, no genotypically representative risk models were identified.

Lastly, regarding the SNP rs2664581, Pearson’s chi-square test revealed no significant relationship between the polymorphism variables and the patient type (*p* > 0.05) (Table 2). Cramer’s V statistic also did not demonstrate a statistically significant degree of association (0.160). In the analysis of the polymorphism’s association with the disease, although models with the polymorphic allele demonstrated an increased risk—particularly the codominant C/A model, which showed a marginal susceptibility to disease risk with an odds ratio (OR) of 1.03 compared to the reference AA genotype—this susceptibility was not substantial enough to be considered a definitive risk model (*p* > 0.05). Furthermore, within the sample size selected for this study, no significant association was observed between the patients’ gender and specific genotypes.

## 3. Discussion

Celiac disease is an autoimmune condition triggered by both genetic and environmental factors. It manifests through an abnormal immune response in the small intestine, resulting in chronic inflammation and several gastrointestinal and extra-intestinal symptoms [1,4,35]. Recent research has shed light on the role of immune checkpoints in the pathogenesis of CD, offering new perspectives on potential therapeutic strategies. The disruption of immune checkpoints in this disease leads to an abnormal immune response to gluten and chronic inflammation in the small intestine [36].

Our previous research demonstrated altered signaling pathways involving the co-inhibitory PD-L1 and immunosuppressive IDO in CD patients compared to healthy individuals [24,25,26]. As such, elevated expression levels of PD-L1 and IDO on the surface of intestinal epithelial and lamina propria cells in patients with CD contribute to the creation of an immunosuppressive environment.

The CD200/CD200R axis is considered an important immunological checkpoint, playing a pivotal role in maintaining immune tolerance and regulating cytokine release [37]. In this study, we characterized, for the first time, the expression of the CD200/CD200R axis in CD using ELISA assay and immunohistochemistry techniques. Abnormalities were found in the expression of elements of the CD200/CD200R pathway in CD patients, revealing an overexpression of both the ligand and the receptor when compared to healthy controls. Our findings demonstrated a significantly higher level of soluble CD200 and CD200R in the serum of CD patients as compared to healthy controls. We propose that the interaction between sCD200 and CD200R is critical for the downstream consequences of the CD200/CD200R axis of immunoregulation. Consistent with our results, CD200 expression is regulated by IFN-gamma in CD and is able to induce CD200 expression, which, in turn, attempts to neutralize the inflammatory response.

The soluble forms of CD200 and CD200R are believed to play a significant role in CD by modulating the immune response and inflammation. The presence of soluble CD200 in the microenvironment is thought to assist in preserving immunological tolerance and restricting autoimmune damage at the intestinal level [29]. Conversely, soluble CD200R forms have the potential to compete with the membrane-bound CD200R in binding to its ligand CD200. This competition may lead to the reduction of inhibitory signaling, potentially prompting a heightened immune response and contributing to disease exacerbation by disrupting normal regulatory mechanisms [29].

Several studies have highlighted the significant role of macrophages in the development and/or progression of CD [38]. Macrophages play a key role in the immune system, acting as a bridge between innate and adaptive immunity, and they are essential participants in the early events that drive CD pathogenesis [38]. During intestinal inflammation, macrophages induce the secretion of pro-inflammatory cytokines such as IFN-γ from activated T cells. Chemokines play a central role in regulating hemopoietic cell movement during the establishment of both inflammation and immune responses [39]. The MIP-4 protein is active on both CD4+ and CD8+ T cells in CD, suggesting that MIP-4 is involved in adaptive rather than innate immunity [40]. The MIP-4 production could be up-regulated by environmental stimuli, such as gluten peptides, thereby prompting distinct functional phenotypes of alternatively activated macrophages.

Macrophages are considered as a primary source of IL23 in the intestine, potentially playing a crucial role in molecular communication with subsets of T cells and innate lymphoid cells within the intestinal environment. This study shows that cells producing IL17 predominantly accumulate in the submucosa and muscularis propria of CD patients, and, as a consequence, elevated levels of IFN-γ and IL17 were observed in comparison to those in healthy controls.

In CD, IFN-γ, primarily secreted by T cells, has the capability to activate tissue transglutaminase 2 (TG2) via the phosphatidylinositol-3-kinase pathway. Conversely, IL-17 and IL-22 serve as regulators, maintaining the integrity of the intestinal mucosa. In CD, the introduction of dietary gluten and the presence of positive TG2 autoantibodies can alter the capacity of intestinal CD4+ and CD8+ T lymphocytes to secrete these cytokines, thereby inducing structural changes in the intestinal tissue. Consequently, both IL-17 and IL-22 may contribute to the development of autoimmune diseases, such as rheumatoid arthritis [41], inflammatory bowel disease [42] and, as observed in our study, in CD.

Elafin, an inducible and multifunctional peptide expressed by mucosal surfaces, including the intestinal epithelium, displays anti-bacterial and anti-inflammatory properties, playing a role in the innate defense during inflammatory events [30]. Acting as an inhibitory protein of serine proteases in epithelial cells, Elafin serves a protective function by mitigating the effects of inflammatory process in immune responses. This includes neutralizing neutrophilic proteases and reducing uncontrolled tissue damage [2].

Galipeau et al. [33] found lower levels of Elafin in patients with active CD, indicative of inflammation and damage to the small intestinal epithelium. Additionally, they described Elafin’s ability to interact with gliadin peptides, impacting TG2 via involvement in their deamidation. In our study, we observed that the expression of Elafin was restricted to the apical surface of the intestinal mucosa, in the lamina propria and on the surface of the Lieberkühn crypts. Lieberkühn crypts, essential for intestinal epithelial cell production, play a crucial role in the gastrointestinal tract. Analyzing Elafin distribution enhances understanding of its protective role against microbial invasion and its involvement in local immune regulation. In celiac disease, Elafin staining’s in Lieberkühn crypts lies in shedding light on its potential role in preserving mucosal integrity amid inflammation and providing insights into the condition’s pathophysiology and immune responses.

We determined Elafin expression in serum of CD and healthy patients. The levels of serum Elafin did not exhibit positive correlations with clinical disease activity (Marsh). Soluble forms of Elafin can act as protease inhibitors in the extracellular environment, thereby limiting tissue degradation and reducing chronic inflammation. In CD, where the balance between proteases and their inhibitors is altered, soluble Elafin forms may help to counteract excessive inflammation and tissue damage.

In celiac patients, during the course of the disease, a robust anti-inflammatory response to gluten may occur. This response often leads to uncontrolled production of Elafin in an attempt to mitigate inflammation. Paradoxically, this excessive production of Elafin can result in the recruitment of intraepithelial lymphocytes, contributing to the persistence of intestinal lesions. Elafin is primarily detected under inflammation conditions, suggesting its inducible nature. Consequently, numerous studies have shown that Elafin expression can be positively regulated in response to several proinflammatory stimuli such as lipopolysaccharide and several proinflammatory cytokines like IL-1β or TNF-α [43].

In this study, we determined the contribution of the *PI3* polymorphism in a cohort of patients diagnosed with active CD and a control group without CD. Various SNPs were selected based on their location in critical regions that are responsible for the translation of the regulatory protein and its inhibitory function. These regions were chosen due to their usual resistance to alternative splicing. Given their presence in exons, significant coding regions, these SNPs underwent genotyping and subsequent analysis to determine their association with CD. The study revealed a notable correlation between the SNP rs1733103 and the risk of CD. The genotype containing the minor polymorphic allele T of rs1733103 is associated with an increased risk, with the overdominant model as the most suitable representation of this association, displaying superior goodness of fit and a lower AIC value. Thus, the presence of the polymorphic allele tentatively indicates an increased risk of altered or diminished activity of *PI3*, contributing to the excessive inflammatory activity in CD.

However, no significant consistency was observed in either of the other two studied polymorphisms. Nevertheless, we could maintain the hypothesis that variations in *PI3* splicing might increase susceptibility to CD, considering the increased expression observed in celiac patients. Further expansion of the sample size is necessary to assert with greater confidence the non-association of these polymorphisms with the alteration of protein activity.

We acknowledge some limitations in our study. While we were able to delineate a series of association models involving the rs1733103 polymorphism significantly with CD, we did not observe a significant association between the rs2664581 and rs41282752 polymorphisms and their influence on CD pathogenesis. However, this lack of replication could be explained by the small sample size within both the cohort of CD patients and the healthy controls, which would require larger sample sizes to either exclude or confirm their contribution. The modification of function associated with alternative splicing variations in association with regulatory genes in autoimmune diseases have already been described [44]. Based on our preliminary results, it is plausible that the post-translational activity of *PI3* could also be influenced by polymorphic activity in important coding regions of the gene.

On the other hand, the absence of a further pathological control group in our study is acknowledged as a potential limitation. While the inclusion of a specific control group could have provided additional insights and strengthened the study’s internal validity, practical constraints and ethical considerations, particularly in the context of working with a pediatric population, made it unfeasible. Our study provides valuable insights for future research toward areas that may benefit from additional investigation.

## 4. Materials and Methods

### 4.1. Study Samples

A total of 96 participants took part in this study, comprising 58 celiac patients and 38 healthy individuals after excluding those with CD. Diagnosis was confirmed via serological tests for tissue transglutaminase (tTG) antibodies and specific HLA-typing for CD. Additionally, biopsy of the small intestine was performed after gastrointestinal endoscopy to validate the diagnosis. Approval from the Ethics Committee was obtained before conducting the study.

### 4.2. Immunohistochemical Staining

Paraffin sections were deparaffinized in xylene, followed by hydration through graded alcohols. Antibodies used in this study included anti-human receptor for the CD200/OX2 cell surface glycoprotein (Abcam, Cambridge, UK), anti-human CD200 (Abcam, Cambridge, UK) and anti-human Elafin/ESI antibody containing 1 WAP domain (Abcam, Cambridge, UK). For antigen retrieval, sections were microwaved in 10 mM citrate buffer (pH 6.0) and subsequently cooled in phosphate-buffered saline (PBS). Sections were then treated with the protein-blocking agent, incubated with the primary antibody, followed by the Specific HRP/DAB IHC Detection Kit—Micropolymer (Abcam, Cambridge, UK). DAB was used as the chromogen. The sections were then counterstained with haematoxylin and mounted with DPX.

For confocal microscopy, the antibodies employed in this study included anti-human IL17 (FabGennix International Inc., Frisco, TX, USA) and anti-human IL22 (LSBio, Seattle, WA, USA), both conjugated with FITC.

To ensure specificity, a negative control section was included, in which the primary antibody was substituted by the corresponding isotype control antibody.

### 4.3. ELISA Assays in Serum from Patients

#### 4.3.1. Analysis of the Levels of Soluble CD200, CD200R and Elafin in Serum

The levels of soluble CD200, CD200R and Elafin in serum of active celiac and non-celiac patients were determined using commercial ELISA kits in accordance with the manufacturer’s instructions (Abcam, Cambridge, UK). For each assay, the standards as well as the samples were tested in duplicate. The cytokine concentrations (pg/mL) were estimated for each assay. The sensitivity of the assays was determined to be >13 pg/mL for sCD200, 12 pg/mL for sCD200R and 1 pg/mL for sElafin, respectively. The concentrations of sCD200, sCD200R and sElafin were quantified from the optical density readings based on standard curves.

#### 4.3.2. Analysis and Expression of Th1 (IFNγ), Th17 (IL23) Cytokine Production

Commercial ELISA kits were used following the manufacturer’s instructions (Thermo Fisher Scientific, Waltham, MA, USA), with standards included on each plate. The sensitivity of the assay for IFNγ was <2 pg/mL and for IL-23 was 4 pg/mL.

#### 4.3.3. Quantitative Measurement of Human MIP4 Levels in the Serum

An in vitro enzyme-linked immunosorbent assay for the quantitative measurement of human MIP4 levels in the serum was used, following the manufacturer’s instructions (Abcam, Cambridge, UK). The minimum detectable dose of MIP4 is typically less than 2 pg/mL.

### 4.4. Elafin Genotype and Allele Frequencies

Genomic DNA was isolated from whole blood samples of the study subjects using the Puregene DNA isolation kit (Gentra Systems, Minneapolis, MN, USA). The genomic DNA was used as a template to generate two PCR products within the coding region of the *PI3* gene. One product was 870 bp in length and contained two SNPs of interest (rs1733103 and rs41282752), while the other product was 463 bp and included the last SNP of interest for this study (rs2664581), corresponding to exons 1 and 2 of *PI3*, respectively. The used primers are listed in Table 3. PCR was performed using DreamTaq Green PCR Master Mix 2× (Thermo Fisher Scientific, Waltham, MA, USA) with 50 ng of genomic DNA and 10 pmol of each primer in a total volume of 25 μL.

The PCR cycling profile consisted of an initial denaturation at 94 °C (2 min), followed by 40 cycles of denaturation at 92 °C (30 s), annealing at 68 °C (30 s) and extension at 72 °C (1 min). A final extension step was performed at 72 °C for 5 min. Aliquots of 5 μL of the PCR product were analyzed by agarose gel electrophoresis (1%) to confirm the correct DNA amplification.

Different aliquots of 20 μL from the PCR reaction product were used for RFLP analysis to genotype the SNPs. For rs1733103 genotyping, 5 units of *Bts*CI (New England Biolabs, UK) were used in a 40 μL reaction mixture, with incubation performed at 37 °C. For rs41282752 genotyping, 5 units of *Bsr*I (New England Biolabs, UK) were used in a 40 μL reaction mixture, with incubation at 65 °C. Finally, for SNP rs2664581 genotyping, 20 μL of the PCR product fragment, along with 5 units of *Hpy*CH4III (New England Biolabs, UK), were incorporated into a 40 μL for digestion, followed by incubation at 37 °C. The results of the digestions (Figure 9) were visualized by electrophoresis on a 4% agarose gel in 0.5× TBE. Genotyping was performed based on the distinct molecular weight banding pattern obtained in each sample.

### 4.5. Statistical Analysis

Statistical analysis of the data was performed using Statgraphics 19 software (StatPoint Technologies, Warrenton, VA, USA). The chi-square test was used for the comparison of qualitative data. The risk association analysis between the polymorphisms and CD was carried out using the SNPstats web tool [45], where the Hardy–Weinberg equilibrium was tested.

## 5. Conclusions

In CD, the dysregulation or compromise of the immune checkpoints contributes to inflammation and the autoimmunity process. CD200 inhibits the immune response by limiting the immune cells activation, while IDO regulates immune tolerance by catabolizing tryptophan. PD1 is a key protein that prevents excessive activation of T lymphocytes. Meanwhile, Elafin modulates the inflammatory response. The study of these checkpoints will lead to the development of targeted therapies aimed at restoring immunological balance in CD. High level of soluble isoforms, such as PD1, Elafin and CD200, are observed in CD and their levels will indicate disease activity and they might be useful in disease monitoring. Soluble isoforms can also be natural competitive inhibitors to membrane-bound isoforms. Therefore, the former block downstream signaling pathways and are potential therapeutic targets.

Our study suggests that alternative splicing of *PI3* might represent another piece in the intricate pathophysiology present in CD patients. We hypothesize a potential differential regulation of *PI3*-splicing variants in CD. Aberrant splicing might result in elevated levels of defective *PI3* protein, which fails to adequately counteract the detrimental effects of cytokines, thus triggering tissue damage associated with autoimmunity. The association study of genotypes derived from the allelic variants of rs1733103 revealed a significant association with increased CD risk. Conversely, we did not observe a notable risk association between the allelic variants rs41282752 SNPs and CD. Although the allelic variants of rs2664581 (C) showed an increased CD risk, statistical significance was not reached. Due to the sample size, the distribution of different genotypes presented in both CD patients and healthy controls did not provide adequate evidence to conclude that these polymorphisms are associated with CD and pose a risk for the disease. Although our findings were inconclusive, there appears to be a trend indicating active polymorphic activity in specific gene regions associated with CD, warranting further investigation in cohorts with larger samples.

In summary, our results corroborate the association between SNPs rs173310 and rs2664581 and an elevated risk of CD, thereby supporting the role of *PI3* in CD development. SNPs in the *PI3* locus may potentially act synergistically, regulating *PI3* gene expression and influencing pre-Elafin biological functions.

## Figures and Tables

**Figure 1 ijms-25-00852-f001:**
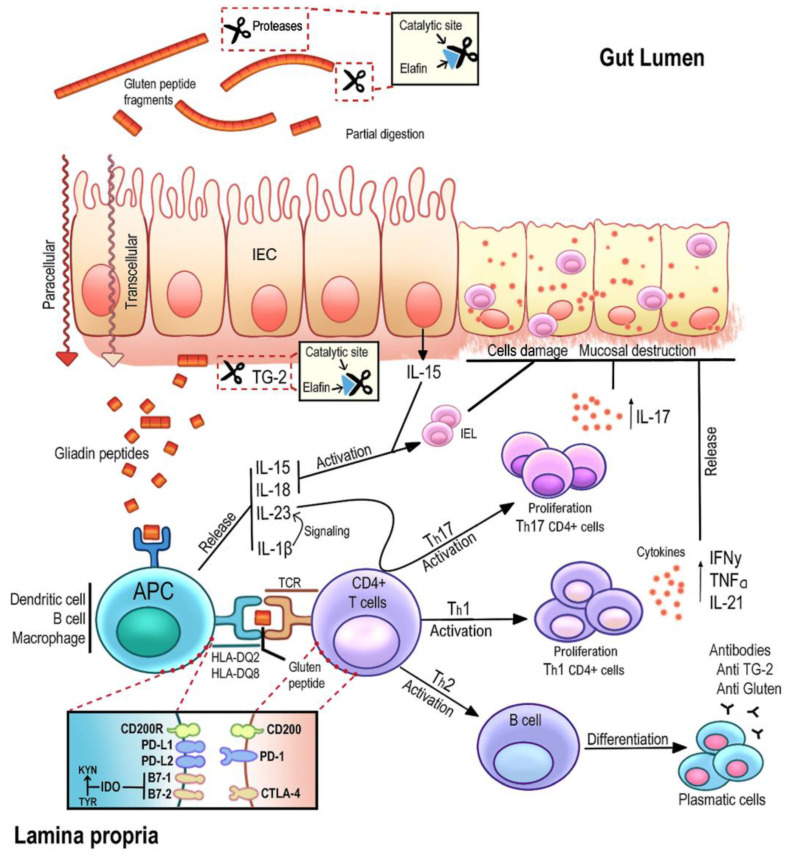
Global schematic representation of the pathogenesis of CD and the area of action of certain immune checkpoint points. Gluten peptides can be transported across the intestinal epithelium. Deamidation by TG2 leads to the production of deamidated gliadin peptides are taken up by lamina propria DCs, inducing a proinflammatory gluten-specific CD4+ T-cell response, characterized by IFN-γ and IL-21 production and anti-gliadin and anti-TG2 antibody production by B cells. Gluten peptides induce epithelial and APC cells to secrete IL-15, resulting in an increase in the number of IELs. Immune checkpoints play an essential role in the function and regulation of effector T cells (Teff) and regulator T cells (Treg). Immune checkpoint molecules limit excessive T cell-mediated inflammatory responses and, in their signaling processes, include a series of ligands that are expressed on the membrane of antigen-presenting cells (APCs), transmitting inhibitory signals. These molecules employ specific receptor partners expressed by T lymphocytes and drive their activation and differentiation or promote immunoregulatory effects. Dysregulation of these signaling processes has been associated with autoimmunity in celiac disease.

**Figure 2 ijms-25-00852-f002:**
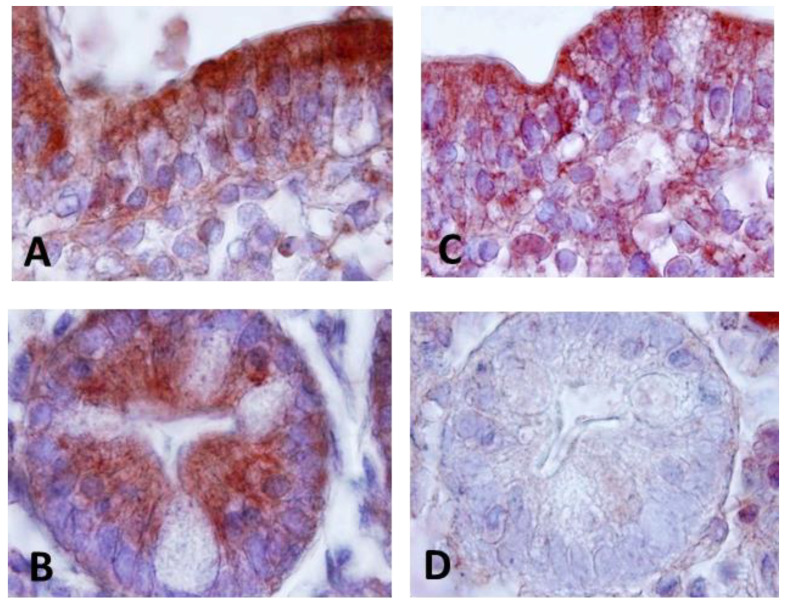
Immunohistochemistry analysis. (**A**) Immunostaining for CD200 in CD patients showing expression in epithelial cells. Magnification 200×. (**B**) Immunostaining for CD200 in Lieberkühn crypts. Magnification 400×. (**C**) Immunostaining for CD200R in CD patients showing expression in epithelial cells and lamina propria cells. Magnification 200×. (**D**) Negative immunostaining for CD200R in Lieberkühn crypts. Magnification 400×.

**Figure 3 ijms-25-00852-f003:**
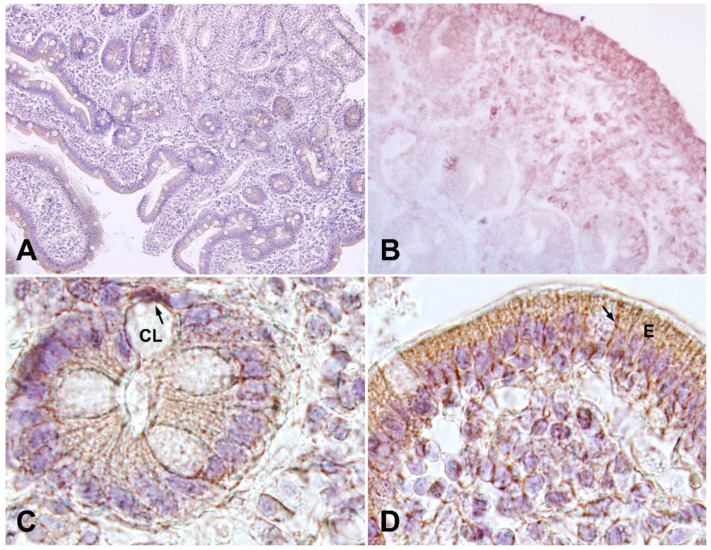
Immunohistochemistry analysis. (**A**) Immunostaining for Elafin in CD patients showing expression in epithelial cells and in Lieberkühn crypts. Magnification 100×. (**B**) Elafin immunostaining in epithelial cells. Magnification 200×. (**C**) Detail of Elafin expression at the level of Lieberkühn crypts (CL). Magnification 400×, (**D**) Detail of Elafin expression in the cells of the epithelium (E) of the intestinal mucosa and lamina propria. Magnification 400×.

**Figure 4 ijms-25-00852-f004:**
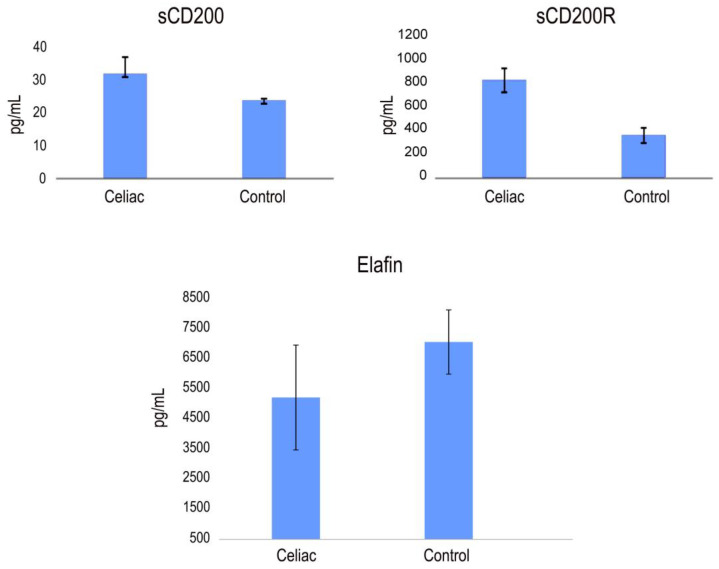
CD200, CD200R and Elafin expression in the serum of CD patients and healthy controls (pg/mL). Significant difference at *p*  <  0.05 is shown.

**Figure 5 ijms-25-00852-f005:**
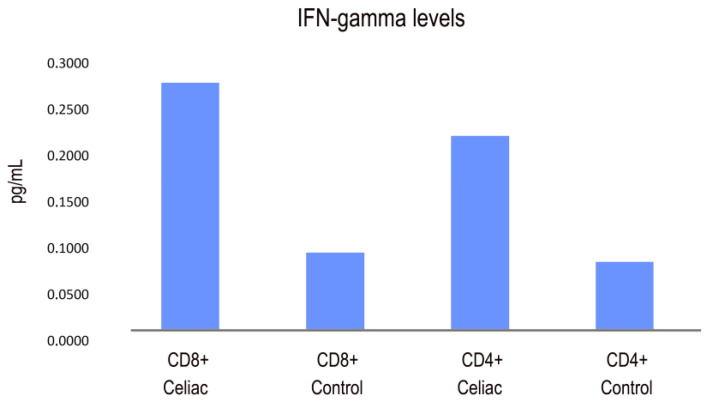
IFN-gamma secretion response in CD and control patients (pg/mL).

**Figure 6 ijms-25-00852-f006:**
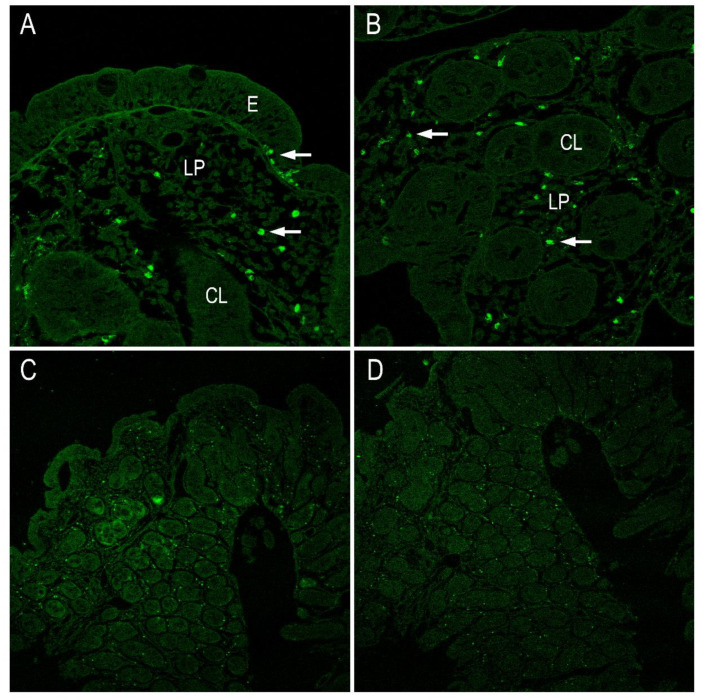
Immunohistochemistry analysis. (**A**) IL17 and (**B**) IL22 in CD patients showing expression in lamina propria and in Lieberkühn crypts (arrows). Magnification 200×. E (epithelium), LP (lamina propria, CL (crypts of Lieberkühn). (**C**) Panoramic image of IL17 expression in the intestinal biopsy at the level of the epithelium, lamina propria and Lieberkühn crypts of the intestinal mucosa. Magnification 100×. (**D**) Panoramic image of IL22 expression in the intestinal biopsy at the level of the epithelium, lamina propria and Lieberkühn crypts of the intestinal mucosa. Magnification 100×.

**Figure 7 ijms-25-00852-f007:**
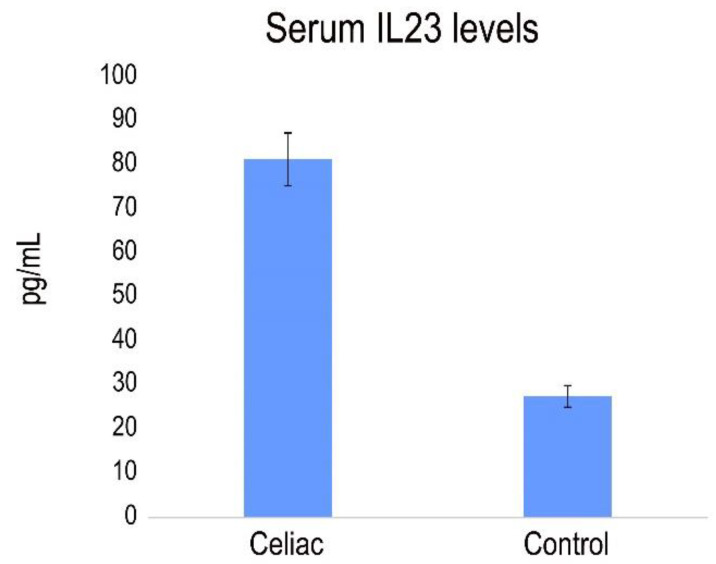
IL23 expression in the serum of CD patients and healthy controls (pg/mL). Significant difference at *p*  <  0.05 is shown.

**Figure 8 ijms-25-00852-f008:**
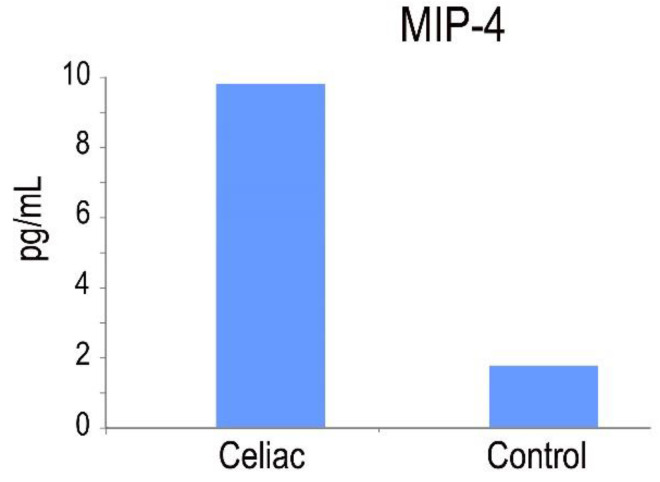
MIP-4 expression in the serum of CD patients and healthy controls (pg/mL). Significant difference at *p*  <  0.05 is shown.

**Figure 9 ijms-25-00852-f009:**
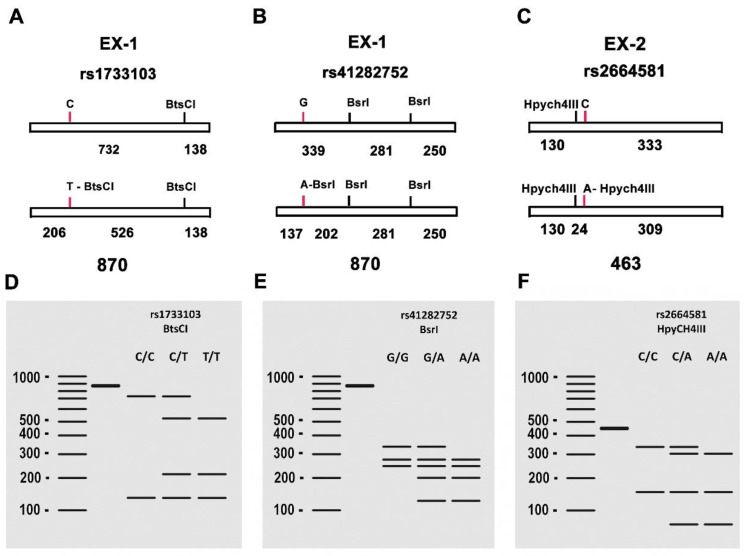
(**A**–**C**) Genotyping of the polymorphisms (C/T), (G/A), (C/A) in the *PI3* gene. (**A**) Schematic representation of the PCR-amplified fragment of 870 bp containing the SNP rs1733103. The presence of T in the SNP creates a recognition site for the *Bts*CI restriction endonuclease, resulting in the generation of three fragments of 138, 206 and 526 bp respectively. (**B**) Schematic representation of the PCR-amplified fragment of 870 bp containing the SNP rs41282752. The presence of A in the SNP creates an additional recognition site for the *Bsr*I restriction endonuclease, resulting in the generation of four fragments of 137, 202, 250 and 281 bp. (**C**) Schematic representation of the PCR-amplified fragment of 463 bp containing the SNP rs2664581. The presence of A in the SNP creates an additional recognition site for the *Hpy*CH4III restriction endonuclease, resulting in the generation of three fragments of 24, 130 and 309 bp. (**D**–**F**) Gel electrophoresis patterns of the PCR products digested with *Bts*CI, *Bsr*I and *Hpy*CH4III, respectively, showing the different banding patterns obtained for rs1733103 (C/C, C/T, T/T), rs41282752 (G/G, G/A, A/A) and rs2664581 (C/C, C/A, A/A).

**Table 1 ijms-25-00852-t001:** Allelic and genotypic frequencies of polymorphic sites in the *PI3* gene in the different study groups.

rs1733103		
Genotypes	Control % (*n*)	Celiac % (*n*)
C/C	74% (28)	48% (28)
C/T	18% (7)	47% (26)
T/T	8% (3)	7% (4)
Alleles	Control	Celiac
C	83% (63)	71% (82)
T	17% (13)	29% (34)
rs41282752		
Genotypes	Control % (*n*)	Celiac % (*n*)
G/G	92% (35)	98% (57)
G/A	8% (3)	2% (1)
G/G	0	0
Alleles	Control	Celiac
G	96% (73)	99% (115)
A	4% (3)	1% (1)
rs41282752		
Genotypes	Control % (*n*)	Celiac % (*n*)
A/A	63% (24)	48% (28)
A/C	29% (11)	45% (26)
C/C	8% (3)	7% (4)
Alleles	Control	Celiac
A	78% (59)	71% (82)
C	22% (17)	29% (34)

**Table 2 ijms-25-00852-t002:** Association between *PI3* genotypes and risk of CD. Abbreviation definitions: CI, confidence interval; OR, odds ratio; AIC, Akaike information criterion; SNP, single nucleotide polymorphism.

rs17333103						
Model	Genotype	Control *n* (%)	Cases *n* (%)	OR (IC 95%)	*p* Value	AIC
Codominant	C/C	28 (73.7%)	28 (48.3%)	1.00	0.022	127.3
	C/T	7 (18.4%)	26 (44.8%)	3.71 (1.39–9.95)		
	T/T	3 (7.9%)	4 (6.9%)	1.33 (0.27–6.51)		
Dominant	C/C	28 (73.7%)	28 (48.3%)	1.00	0.012	126.6
	C/T + T/T	10 (26.2%)	30 (51.7%)	3.00 (1.24–7.28)		
Recessive	C/C + C/T	35 (92.1%)	54 (93.1%)	1.00	0.85	132.9
	T/T	3 (7.9%)	4 (6.9%)	0.86 (0.18–4.10)		
Overdominant	C/C + T/T	31 (81.6%)	32 (55.2%)	1.00	0.0063	125.4
	T/T	7 (18.4%)	26 (44.8%)	3.60 (1.36–9.49)		
Log-Additive				1.94 (0.95–3.97)	0.057	129.3
rs41282752						
Model	Genotype	Control *n* (%)	Cases *n* (%)	OR (IC 95%)	*p* Value	AIC
	G/G	35 (92.1%)	57 (98.3%)	1.00	0.14	130.7
	G/G	3 (7.9%)	1 (1.7%)	0.20 (0.02–2.05)		
rs2664581						
Model	Genotype	Control *n* (%)	Cases *n* (%)	OR (IC 95%)	*p* Value	AIC
Codominant	A/A	24 (63.2%)	28 (48.3%)	1.00	0.28	132.4
	C/A	11 (28.9%)	26 (44.8%)	2.03 (0.83–4.94)		
	C/C	3 (7.9%)	4 (6.9%)	1.14 (0.23–5.62)		
Dominant	AA	24 (63.2%)	28 (48.3%)	1.00	0.15	130.8
	C/A + A/A	14 (36.8%)	30 (51.7%)	1.84 (0.80–4.24)		
Recessive	A/A + C/A	35 (92.1%)	54 (93.1%)	1.00	0.85	132.9
	C/C	3 (7.9%)	4 (6.9%)	0.86 (0.18–4.10)		
Overdominant	A/A + C/C	27 (71%)	32 (55.2%)	1.00	0.11	130.4
	C/A	11 (28.9%)	26 (44.8)	1.99 (0.83–4.77)		
Log-Additive				1.44 (0.73–2.82)	0.29	131.7

**Table 3 ijms-25-00852-t003:** Primer sequences and fragments generated by PCR. The restriction endonuclease used for the analysis of each SNP is also indicated.

Region	Primer Code	Sequence	PCR Product (bp)	SNP	Restriction Endonuclease
Exon 1	PI3-119-F	CCCAGGTCCCTCCCAGAA	870	rs1733103	*Bts*CI
	PI3-969-R	CCTTCCTCCACTCCAAGTCT		rs41282752	*Bsr*I
Exon 2	PI3-1080-F	CTTCCCTACTCAGGCCATGG	463	rs2664581	*Hpy*CH4III
	PI3-1523-R	CGCTCAGCCTTCTTTTGTGT			

## Data Availability

The raw data supporting the conclusions of this article will be made available by the authors, without undue reservation.

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
