# Peer review of "Expression of Elafin and CD200 as Immune Checkpoint Molecules Involved in Celiac Disease"

_ijms, 2024, doi:10.3390/ijms25020852_

Round 1
Reviewer 1 Report
Comments and Suggestions for Authors
In this study, Candelaria Ponce-de-León and colleagues, aimed to characterize the expression and function of the CD200/CD200R axis and
peptidase inhibitor 3 (PI3) in Celiac Diseas
e (CD), along with their correlation with other immune checkpoints, such as IDO and PD1/PDL1. They also investigated whether the interaction of CD200/CD200R reduces activation, proliferation, and Th17 cytokine production. Overall, they studied 58 CD patients and 38 healthy individuals. CD200, CD200R and Elafin distribution were assessed by ELISA and immunohistochemistry analyses in serum and biopsies of CD patients, as well as analyses of Th1 and Th17 cytokines. PCR amplification of a fragment of the PI3 gene was also performed by using genomic DNA isolated from whole blood samples. They found a significantly higher level of soluble CD200, CD200R and lower expression of PI3 in serum of CD patients, observed compared to healthy controls. Regarding the PI3 genotype, the CT genotype proportion SNP rs1733103 and the GG genotype SNP rs41282752, were predominant in CD patients. SNP rs1733103 showed a significant association between the SNP variables and CD. They concluded that in CD the immune checkpoint is compromised or dysregulated, and this can contribute to inflammation and the autoimmunity process thus supporting the development of targeted therapies aimed at restoring immunological balance in CD to modify the Elafin protein expression and function.
The study is of interest since explores novel ways to better understand the pathogenetic mechanisms underlying the mucosal inflammation. I have minor points requiring to be addressed.
1) Study population details: additional information related to the CD diagnosis should be provided. Tissue transglutaminase: could the authors provide additional informations (commercial (which one?) or in-house ELISA test?). Duodenal histology: how was assessed? Marsh-Oberhuber criteria?
2) In my opinion, to further sustain the CD-specificity of the study findings, a pathologic control group (i.e. non-celiac intestinal disease) would be appropriate. If there no further pathological control group, this should be recognized as a potential limit of the study.
3) Introduction: the authors discuss the important role of TG2 in CD pathogenesis by deamidating gluten peptides and increasing the binding affinity of HLA-DQ2 (DQ8) molecules to these peptides. In this regard, they also should recall the important role of TG2 in the so-called secondary autoimmunity.In fact, by deamidating target proteins, TG2 may lead to neo-antigen production and autoantibody/autoimmunity generation as previously described (Dig Liver Dis. 2002 Jan;34(1):13-5. doi: 10.1016/s1590-8658(02)80053-6.). This may explain the frequent finding of serum autoantibodies associated with extraintestinal immune-mediated manifestations in CD patients such as anti-ganglioside antibodies in CD patients with neurological disorders as previously described (doi: 10.1186/s13223-021-00557-y. PMID: 34049567), as well as anti-actin IgA antibodies in CD patients with severe mucosal damage as demonstrated (doi: 10.1111/j.1365-2249.2004.02541.x. PMID: 15270857), and in CD patients with atopy, as reported (doi: 10.1016/s1590-8658(00)80354-0. PMID: 11215557.).
Author Response
REVIEWER 1. Thank you for your interesting suggestions regarding our study that Undoubtedly, they have significantly improved our manuscript
- Study population details: additional information related to the CD diagnosis should be provided. Tissue transglutaminase: could the authors provide additional informations (commercial (which one?) or in-house ELISA test?). Duodenal histology: how was assessed? Marsh-Oberhuber criteria
- Reply: In response to your first point on study population details, we will indeed include a supplementary table (suplemmentary table 1) with additional information related to the celiac disease (CD) patients. In all cases, diagnosis of CD was confirmed by endoscopic duodenal biopsy, revealing different grades of villous atrophy according to the modified Marsh classification). In all CD patients, intestinal villous atrophy was associated with a positivity for serological CD markers (anti-endomysial and/or antitissue transglutaminase antibodies) further supporting the diagnosis of CD. All these available data, were obtained from the hospital digital database
- In my opinion, to further sustain the CD-specificity of the study findings, a pathologic control group (i.e. non-celiac intestinal disease) would be appropriate. If there no further pathological control group, this should be recognized as a potential limit of the study
- Reply: Including a pathologic control group with non-celiac intestinal disease to compare and contrast the findings specific to celiac disease can help in identifying unique characteristics or markers associated with CD. In our study to include such a control group focuses on a children's population. Research involving children requires careful ethical considerations, and exposing pediatric participants to unnecessary invasive procedures or interventions solely for the purpose of creating a control group may raise ethical concerns. It's crucial to prioritize the well-being and safety of child participants. Adding a pathologic control group with non-celiac intestinal disease may introduce complexities that are challenging to navigate in a pediatric setting. This limitations of a research are raised in the discussion section. In the text we have include the followign text “The absence of a further pathological control group in our study is acknowledged as a potential limitation. While the inclusion of a specific control group could have provided additional insights and strengthened the study's internal validity, practical constraints and ethical considerations, particularly in the context of working with a pediatric population, made it unfeasible. Our study provides valuable insights for future research toward areas that may benefit from additional investigation”.
- Introduction: the authors discuss the important role of TG2 in CD pathogenesis by deamidating gluten peptides and increasing the binding affinity of HLA-DQ2 (DQ8) molecules to these peptides. In this regard, they also should recall the important role of TG2 in the so-called secondary autoimmunity.In fact, by deamidating target proteins, TG2 may lead to neo-antigen production and autoantibody/autoimmunity generation as previously described (Dig Liver Dis. 2002 Jan;34(1):13-5. doi: 10.1016/s1590-8658(02)80053-6.). This may explain the frequent finding of serum autoantibodies associated with extraintestinal immune-mediated manifestations in CD patients such as anti-ganglioside antibodies in CD patients with neurological disorders as previously described (doi: 10.1186/s13223-021-00557-y. PMID: 34049567), as well as anti-actin IgA antibodies in CD patients with severe mucosal damage as demonstrated (doi: 10.1111/j.1365-2249.2004.02541.x. PMID: 15270857), and in CD patients with atopy, as reported (doi: 10.1016/s1590-8658(00)80354-0. PMID: 11215557.).
- Reply: In introduction we have included the following text and in references we have added the works cited. “Transglutaminase 2 (TG2) is crucial in secondary autoimmunity, generating neo-antigens and triggering autoantibody production in celiac disease. This process leads to the presence of serum autoantibodies, including anti-ganglioside and anti-actin IgA antibodies, in CD patients with neurological disorders or severe mucosal damage. TG2's role sheds light on extraintestinal immune manifestations, revealing the intricate interplay between TG2, autoimmunity, and diverse clinical presentations in CD patients”

Reviewer 2 Report
Comments and Suggestions for Authors
This study focused on the analysis of CD200 and CD200R and PI3 (elafin) in celiac disease. The analysis is made in biopsies, and additional work is using ELISA in serum. Alterations of these markers were found in the patients. The study point out the CD200-CD200R axis an important. Although what cells express these markers if not fully clarified.
Comment:
(1) In the Introduction. Could you please make a figure showing the pathogenesis of celiac disease, the cells of the microenvironment, and the expression of the CD200 and CD200R? This may help the reader to identify why CD200-CD200R axis is important.
(2) Please refer to this manuscript.
Chen EY, Chu S, Gov L, Kim YK, Lodoen MB, Tenner AJ, Liu WF. CD200 modulates macrophage cytokine secretion and phagocytosis in response to poly(lactic co-glycolic acid) microparticles and films. J Mater Chem B. 2017 Feb 28;5(8):1574-1584. doi: 10.1039/C6TB02269C. Epub 2017 Jan 10. PMID: 28736613; PMCID: PMC5515357. This manuscript points out to macrophages. You may explain the M1/M2 macrophage polarization. (3) Could you please expand the description of the pleiotropic mechanisms of cation of the Elafin/trappin-2 pathway? Please refer to: Deraison C, Bonnart C, Langella P, Roget K, Vergnolle N. Elafin and its precursor trappin-2: What is their therapeutic potential for intestinal diseases?. Br J Pharmacol. 2023;180(2):144-160. doi:10.1111/bph.15985 (4) Figure 3 shows the serum expression of CD200, CD200R, and Elafin. Is it possible to add the detailed p values in the graphs? In serum, what cells express and/or secrete these molecules? (5) Is it possible to have additional images of the immunohistochemical analyses? In Figure 1, in the current magnification it is difficult to see what cells are expressing the markers. (6) In figure 2. In the current magnification the expression of Elafin in CD of Figure 2A looks negative. Could you please provide additional images? (7) What is the importance of the staining in Lieberkühn crypt levels? (8) Regarding "When isotype controls were used, there was an absence of immune reactivity patterns (data not shown)". Could you please show the data (if available)? (9) In section 2.1. Is it possible to add a table, for example in the appendix, with the clinicopathological characteristics of the patients? (10) Figure 4 is not necessary to be in 3D as the third dimension is not providing any additional information. Are the results from circulating CD4 and CD8-positive cells? Are these cells the same as in the intestinal mucosa? (11) In Figure 5, positive cells are seen, but the staining of the crypts seems missing. (12) In Figure 5, are the IL17+cells, of the type of Th17? (13) In the Elisa of Figure 7. Why MIP-4 is expressed by macrophages? Did you proved it? (14) Regarding "Notably, in this study, the observed genotype frequencies in the control and CD groups were consistent with the Hardy-Weinberg equilibrium". Could you please describe this equilibrium, and the interpretation? (15) Is it possible to add the p values in Table 1? Is it feasible to add the strenght of the association? (16) All all the polymophisms associated exclusively to the PI3 gene? (17) Regarding "Consistent with our results, CD200 expression is regulated by IFN-gamma. Interaction of CD200/CD200R leads to production of type Th1 and Th17 cytokines." Could you please show where this data is shown? (18) In the abstract, regarding the conclusion "In celiac disease the immune checkpoint is compromised or dysregulated, which can contribute to inflammation and the autoimmunity process.". Please confirm that the data and results of this study confirm this statement. (19) Regarding "Specific coding regions of the PI3 gene splice variants predispose the Elafin protein, both at the transcriptional and post-translational levels, to modify its expression and function, resulting in reduced differential functional protein levels in patients with active celiac disease." Sorry to ask, but, how did you prove it? (20) Please add the catalog number of the primary antibodies, and other necessary reagents. (21) Figure 8. Please confirm that this figure has no mistake. (22) What about the data of PD-L1 and PD-1. Was it evaluated?
Author Response
REVIEWER 2. Thank you for your interesting suggestions and comments regarding our study that Undoubtedly, they have significantly improved our manuscript
1.In the Introduction. Could you please make a figure showing the pathogenesis of celiac disease, the cells of the microenvironment, and the expression of the CD200 and CD200R? This may help the reader to identify why CD200-CD200R axis is important
- In the introduction we have included a new figure (Figure 1) showing the pathogenesis of celiac disease, the cells of the microenvironment, and the expression of the CD200 and CD200R
- Please refer to this manuscript.: Chen EY, Chu S, Gov L, Kim YK, Lodoen MB, Tenner AJ, Liu WF. CD200 modulates macrophage cytokine secretion and phagocytosis in response to poly(lactic co-glycolic acid) microparticles and films. J Mater Chem B. 2017 Feb 28;5(8):1574-1584.
- Reply: We have refered this work and we have explained the M1/M2 macrophage polarization and CD200 in the introduction chapter
- (3) Could you please expand the description of the pleiotropic mechanisms of cation of the Elafin/trappin-2 pathway? Please refer to: Deraison C, Bonnart C, Langella P, Roget K, Vergnolle N. Elafin and its precursor trappin-2: What is their therapeutic potential for intestinal diseases?. Br J Pharmacol. 2023;180(2):144-160. doi:10.1111/bph.15985
- Reply: We have expanded the description of the pleiotropic mechanisms of cation of the Elafin/trappin-2 pathway and their therapeutic potential for intestinal diseases in introduction chapter. We have include this new reference
- Figure 3 shows the serum expression of CD200, CD200R, and Elafin. Is it possible to add the detailed p values in the graphs? In serum, what cells express and/or secrete these molecules?
- Reply: We have added the p values in Figure 3. In relation with the second question Soluble forms of CD200 and CD200R, detected in bodily fluids including serum, result from processes like shedding or alternative splicing. Their detection in serum is context-dependent and varies in physiological or pathological conditions. While Elafin is primarily linked to mucosal defense, its presence in serum, particularly during inflammation or diseases, suggests it could serve as a potential biomarker. In the context of celiac disease, soluble immune checkpoints are often detected in serum, indicating their release through alternative splicing, shedding, or active secretion.
- Is it possible to have additional images of the immunohistochemical analyses? In Figure 1, in the current magnification it is difficult to see what cells are expressing the markers.
- Reply: If the size is increased further in Figure 1 (now Figurw 2), the field of view of the tissue sample is lost, and thus, the degree of protein and receptor expression cannot be observed. Including images C and D with double the magnification already allows for better appreciation of the details. The figure legends explain the type of cells expressing CD200 and CD200R
- In figure 2. In the current magnification the expression of Elafin in CD of Figure 2A looks negative. Could you please provide additional images?
- Reply: In figure 2 (now Figure 3) we have included additional images a high magnification to better observe the expression of Elafin at the level of the epithelium and Lieberkühn crypts
- What is the importance of the staining in Lieberkühn crypt levels?
- Reply: Lieberkühn crypts, essential for intestinal epithelial cell production, play a crucial role in the gastrointestinal tract. Analyzing Elafin distribution enhances understanding of its protective role against microbial invasion and its involvement in local immune regulation. In celiac disease, Elafin staining's in Lieberkühn crypts lies in shedding light on its potential role in preserving mucosal integrity amid inflammation and providing insights into the condition's pathophysiology and immune responses. We have included this paragraph in the discusión chapter
- Regarding "When isotype controls were used, there was an absence of immune reactivity patterns (data not shown)". Could you please show the data (if available)?
- Reply: Since isotype controls do not exhibit specific binding to the target of interest, these data may not be visually presented to avoid cluttering figures with non-informative signals. Now, I am including one of the images with absence immune reactivity, that is not included in the manuscript.
- In section 2.1. Is it possible to add a table, for example in the appendix, with the clinicopathological characteristics of the patients?
- Reply: I have incuded as supplementary table 1with the clinical charactersitics of celiac patients
- Figure 4 is not necessary to be in 3D as the third dimension is not providing any additional information. Are the results from circulating CD4 and CD8-positive cells? Are these cells the same as in the intestinal mucosa?
- Reply: In figure 4 (now Figure 5) has been included the corrected version. In relation with the second question Although circulating CD4 and CD8-positive cells in the serum share common cell surface markers with those present in the intestinal mucosa, their functional characteristics and roles can be shaped by the specific microenvironment in which they reside. Immune cells in the intestinal mucosa are adapted to the unique challenges and functions of the gut, and these adaptations can result in differences between circulating and mucosal subsets. The intestinal mucosa has a distinct immune system with specialized immune cells that contribute to maintaining tolerance to food antigens, interact with the intestinal microbiota, and provide defense against pathogens.
- In Figure 5, positive cells are seen, but the staining of the crypts seems missing.
- Reply: We have included a new 2 images at low maginification in figure 5 (now Figure 6) showing positive stainign cells of the crypts.
- In Figure 5, are the IL17+cells, of the type of Th17?
- Reply: Th17 cells are a subset of T helper cells major producers IL-17 playing a crucial role in immune responses and are implicated in autoimmune diseases. The differentiation and function of Th17 cells are tightly regulated by the cytokine such as IL-23, IL-1beta, and IL-21 are also involved in maintaining Th17 cell function
- In the Elisa of Figure 7. Why MIP-4 is expressed by macrophages? Did you proved it?
- Reply: MIP-4 (Macrophage Inflammatory Protein 4) is indeed expressed by macrophages playing a central role in the innate immune system and the regulation of immune responses. By expressing MIP-4, macrophages can recruit other immune cells to the site of inflammation and may modulate the immune response, contributing to the balance between pro-inflammatory and anti-inflammatory signals. Research on MIP-4 and macrophages may have implications for celiac disease diseases where immune dysregulation and inflammation are contributing factors
- Regarding "Notably, in this study, the observed genotype frequencies in the control and CD groups were consistent with the Hardy-Weinberg equilibrium". Could you please describe this equilibrium, and the interpretation?
- Reply: Hardy-Weinberg equilibrium is a principle in population genetics that describes the expected genotypic frequencies in an ideal population that meets specific conditions. These conditions include an infinite population size, absence of migration, absence of mutations, absence of natural selection and random mating. Under these conditions, allele and genotypic frequencies in a population remain constant from generation to generation. Hardy-Weinberg equilibrium can be a useful tool in the study of the relationship between a genetic polymorphism and a disease. If the genotypic frequencies observed in the population affected by the disease differ significantly from the expected genotypic frequencies according to Hardy-Weinberg equilibrium in the control population, it could indicate a strong association between the studied polymorphism and the disease. This disparity could indicate that the polymorphism may be related to an increased risk or susceptibility to the disease, or even have a causal role in it.
- Is it possible to add the p values in Table 1? Is it feasible to add the strenght of the association?
- Reply: The p values are included in the text amd the strenght of the association área included in table 2
- All all the polymophisms associated exclusively to the PI3 gene?
- Reply: The study is the genotypes distributions of rs17333103, rs41282752 (both in Exón 1) (Exón 1) and rs2664581 (Exón 2) in PI3gene
- Regarding "Consistent with our results, CD200 expression is regulated by IFN-gamma. Interaction of CD200/CD200R leads to production of type Th1 and Th17 cytokines." Could you please show where this data is shown?
- Reply: The regulation of CD200 expression by interferon-gamma and the subsequent interaction of CD200 with its receptor CD200R influencing the production of type Th1 and Th17 cytokines are supported by various studies in immunology. IFN-gamma has been shown to upregulate the expression of CD200 on different cell types. While CD200/CD200R interaction generally exerts inhibitory effects, some studies suggest that in specific contexts, it may contribute to the polarization of T cells towards the Th1 and Th17 phenotypes. Th1 and Th17 cells are associated with pro-inflammatory responses. The exact mechanisms are not fully elucidated, but it's thought that the inhibitory signals from CD200/CD200R interaction may selectively affect certain T cell subsets. The dysregulation of CD200/CD200R interaction has been implicated in other autoimmune diseases and inflammatory conditions and with our work, also is observed for the first time in celiac disease
- In the abstract, regarding the conclusion "In celiac disease the immune checkpoint is compromised or dysregulated, which can contribute to inflammation and the autoimmunity process.". Please confirm that the data and results of this study confirm this statement.
- Reply: The study reveals that in celiac disease, there is dysregulation or compromise of immune checkpoints, disrupting normal immune tolerance mechanisms and leading to an inappropriate and sustained immune response against gluten. This dysregulation contributes to inflammation, tissue damage, and the autoimmune nature of celiac disease. High levels of soluble isoforms, including Elafin and CD200, are observed in celiac disease, indicating disease activity and potential utility in disease monitoring. Abnormalities in the CD200/CD200R pathway show overexpression of both the ligand and receptor in celiac patients compared to healthy controls. The study finds significantly higher levels of soluble CD200 and CD200R in the serum of celiac patients. Soluble CD200R has the potential to compete with the membrane-bound form, reducing inhibitory signaling and potentially heightening the immune response, contributing to disease exacerbation. In celiac patients, an anti-inflammatory response to gluten may lead to excessive Elafin production, paradoxically contributing to the persistence of intestinal lesions by recruiting intraepithelial lymphocytes.
- Regarding "Specific coding regions of the PI3 gene splice variants predispose the Elafin protein, both at the transcriptional and post-translational levels, to modify its expression and function, resulting in reduced differential functional protein levels in patients with active celiac disease." Sorry to ask, but, how did you prove it?
- Reply: By employing a combination of two experimental approaches, we can build a comprehensive understanding of how specific coding regions of PI3 gene splice variants impact Elafin expression and function in the context of celiac disease, providing robust evidence to support the initial statement. Conduct genetic analyses, such as DNA genotyping, to identify and characterize the specific coding regions of PI3 gene splice variants. This can reveal variations or mutations associated with celiac disease. Protein Expression Analysis, utilize enzyme-linked immunosorbent assay (ELISA) to measure Elafin protein levels in intestinal tissue associated with different PI3 gene variants can provide insights into post-translational modifications affecting protein expression.
- Please add the catalog number of the primary antibodies, and other necessary reagents.
- Reply: We have included in methods section the reference of the antibodies and other reagents
- Figure 8. Please confirm that this figure has no mistake.
- Reply: We have corrected the mistake in figure 8
- What about the data of PD-L1 and PD-1. Was it evaluated?
- Reply: The data of PD-L1 and PD-1 were previously evaluated bu our research group and published in Cell Mol Immunol 2019 doi: 10.1038/s41423-019-0256-7, and Frontiers Immunology 2021, doi: 10.3389/fimmu.2021.678400. These references are cited in the work

Round 2
Reviewer 1 Report
Comments and Suggestions for Authors
In the revised manuscript the authors addressed thè raised points and the manuscript can be accepted.